# Emergent periodicity in the collective synchronous flashing of fireflies

Raphael Sarfati[1]*[†], Kunaal Joshi[2]*[†], Owen Martin[1][†], Julie C Hayes[3], Srividya Iyer-Biswas[2,4]*, Orit Peleg[1,4]*

[1]BioFrontiers Institute, University of Colorado Boulder, Boulder, United States; [2]Department of Physics and Astronomy, Purdue University, West Lafayette, United States; [3]Department of Computer Science, University of New Mexico, Albuquerque, United States; [4]Santa Fe Institute, Santa Fe, United States

**Abstract** In isolation from their peers, *Photinus carolinus* fireflies flash with no intrinsic period between successive bursts. Yet, when congregating into large mating swarms, these fireflies transition into predictability, synchronizing with their neighbors with a rhythmic periodicity. Here we propose a mechanism for emergence of synchrony and periodicity and formulate the principle in a mathematical framework. Remarkably, with no fitting parameters, analytic predictions from this simple principle and framework agree strikingly well with data. Next, we add further sophistication to the framework using a computational approach featuring groups of random oscillators via integrate-and-fire interactions controlled by a tunable parameter. This agent-based framework of *P. carolinus* fireflies interacting in swarms of increasing density also shows quantitatively similar phenomenology and reduces to the analytic framework in the appropriate limit of the tunable coupling strength. We discuss our findings and note that the resulting dynamics follow the style of a decentralized follow-the-leader synchronization, where any of the randomly flashing individuals may take the role of the leader of any subsequent synchronized flash burst.

**\*For correspondence:**
raphael.sarfati@cornell.edu (RS);
joshi84@purdue.edu (KJ);
iyerbiswas@purdue.edu (SI-B);
orit.peleg@colorado.edu (OP)

[†]These authors contributed equally to this work

**Competing interest:** The authors declare that no competing interests exist.

## Editor's evaluation

This important study provides a quantitative characterization and understanding of firing collective patterns in P. Carolinus fireflies. The work significantly contributes to fill the gap between observations and mechanistic models, with convincing experimental evidence and solid theoretical modeling. This work will be of interest to readers curious about collective behavior, biological rhythms, and models of synchronized oscillations.

## Introduction

Physical systems consisting of several interacting entities often exhibit large-scale properties which are distinct from the capabilities of each entity taken individually: this is the well-known concept of emergence. Emergence has been observed and studied in both inanimate and animate systems, including famously groups of animals (**Kelley and Ouellette, 2013**; **Attanasi et al., 2014**). Animal collective behavior broadly designates dynamical patterns that are unsupervised consequences of the accumulation of low-level interactions between neighboring individuals (**Ouellette, 2022**; **Ballerini et al., 2008**; **Couzin, 2009**). One simple yet compelling manifestation of emergence in the natural world is in the form of firefly flash synchronization (**Faust, 2010**; **Buck and Buck, 1966**; **Sarfati et al., 2020**; **Sarfati et al., 2021**; **Sarfati et al., 2022**). For example, when sufficiently many *Photinus carolinus* fireflies congregate into a mating swarm (lek), they start to align their flashes on the same tempo, creating a mesmerizing display that has captivated the curious minds of many. This possibly serves to

strengthen their species-specific signal and heighten the ability for conspecific males and females to identify one another (*Faust, 2010*; *Moiseff and Copeland, 2010*; *Stanger-Hall and Lloyd, 2015*). In addition to collective synchrony, a more careful examination of *P. carolinus'* flashing pattern further reveals another non-trivial signature: emergent *periodicity*. Indeed, in their natural habitat, these fireflies produce periodic bursts of flashes occurring with great regularity, with a temperature-dependent period generally around 12 s (*Faust, 2010*; *Moiseff and Copeland, 2010*). Surprisingly, when put in isolation, a single firefly does not appear to show any regularity about when it emits its flash trains (*Sarfati et al., 2020*), where intervals between flash trains vary between a few seconds to a few minutes apart. How, then, can a multitude of interacting fireflies exhibit a specific frequency that does not appear to be encoded in any single one of them?

Synchronization is traditionally thought of as the adjustment of rhythms of self-sustained oscillators due to coupling (*Mirollo and Strogatz, 1990*; *Strogatz, 1997*; *Strogatz, 2000*; *Pikovsky et al., 2001*; *Ramírez Ávila et al., 2003*; *Ramírez Ávila et al., 2019*). The Kuramoto model and other such traditional models addressing synchrony in systems such as *Pteroptyx malaccae* fireflies model individuals as oscillators firing highly regularly in isolation, often with different periods. The question these models are primed to answer is, how do oscillators with different individual periods and starting from different phases, come together to oscillate synchronously? This is fundamentally different from the problem posed by our system, in which the individuals, which fire highly irregularly, seem to use synchrony through coupling with other individuals as a tool to achieve greater regularity in their firing period. While traditional models of limit-cycle oscillators are also capable of modeling systems in

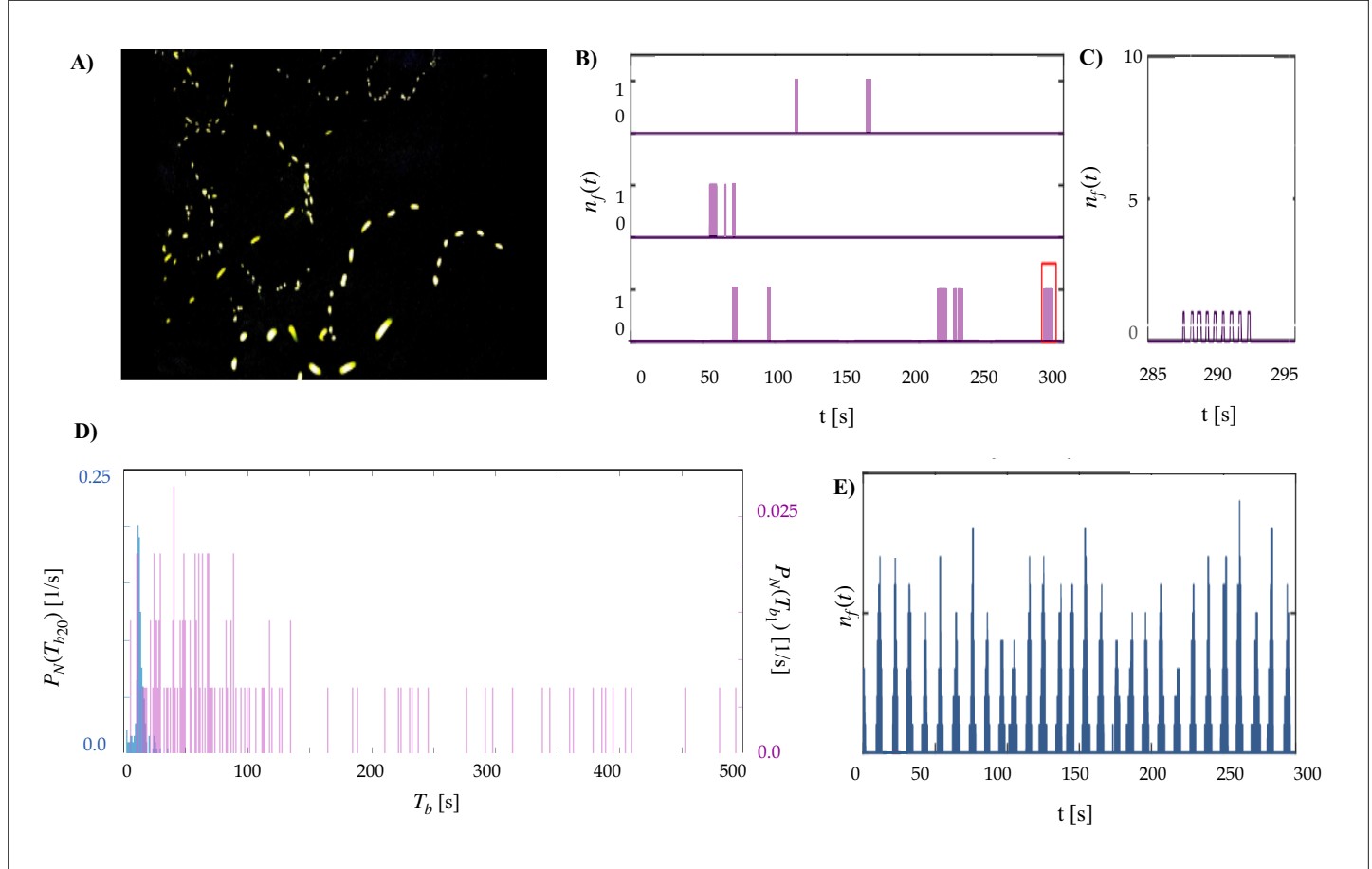

**Figure 1.** Schematic representation of the proposed principle and its theoretical implication: the emergence of periodicity in stochastically flashing *P. carolinus* fireflies. (**A**) Long exposure photograph illustrating flashes in a *P. carolinus* natural swarm. (**B**) Overlaid time series of three isolated individual fireflies emitting flash bursts which appear random. The inset (**C**) shows the burst-like nature of *P. carolinus* flash events. (**D**) Interburst distributions $b(t)$ for one firefly (purple) and 20 fireflies (blue) insulated from the rest of the swarm. (**E**) Twenty *P. carolinus* fireflies flashing in a tent exhibiting the periodic nature of their collective flashing.

which isolated individuals do not oscillate periodically but collective oscillations occur only above a certain threshold density, in those models the individuals are generally inherently oscillatory and their periodic oscillations are suppressed through a sufficiently strong coupling with the surroundings. (*De Monte et al., 2007*; *Taylor et al., 2009*). We instead present a stochastic theoretical framework based on a simple, intuitive mechanism by which inherently non-oscillating individuals are able to oscillate synchronously in a group, and apply this to *P. carolinus* fireflies, successfully explaining the convergence towards a common, well-defined period between flash bursts as the number $N$ of fireflies increases.

## Results
### Behavioral experiments

A *P. carolinus* lek in its natural habitat contains several thousands of fireflies of which the males display a robust collective flash pattern. They flash over the course of periodic bursts separated by a few seconds of total darkness (*Figure 1A*, over a few seconds). Collective bursts in the swarm have a well-defined period (peak-to-peak) of about 12 s (*Sarfati et al., 2020*). One could think, then, that each individual firefly also emits flash trains with about the same time period, and that the effect of visual interactions is to align these individual trains on the same tempo. In other words, the swarm could be a set of coupled oscillators converging to a common phase, as has been described in previous models (*Mirollo and Strogatz, 1990*; *Strogatz, 1997*; *Strogatz, 2000*; *Ermentrout, 1991*; *Rodrigues et al., 2016*). Crucially, however, when a single firefly is taken out of the lek and placed in a large (2 m³) enclosing volume visually insulated from the rest of the group, all periodicity in the occurrence of flash trains is lost. The single firefly continues to emit sporadic bursts (*Figure 1B and C*), but the time between successive flash bursts varies between a few seconds and a few minutes (*Figure 1B* and *Sarfati et al., 2020*). This suggests that individual interburst intervals (IBIs) occur at random, and may thus depend on a variety of behavioral factors. When collecting measurements from 10 different fireflies recorded for several minutes under the same conditions, we are able to outline the distribution of interburst intervals for a single firefly in isolation (*Figure 1D*, purple). (The underlying assumption here is that all fireflies have the same distribution of interburst intervals.) Interestingly, as the number of fireflies within the enclosing volume is increased, a regularity in the time between bursts starts to emerge. At about $N = 15$, the distribution of interburst intervals becomes very similar to that observed in the natural habitat (*Sarfati et al., 2020*). For $N = 20$, it is clear that there is a very strong collective periodicity in the emission of flash bursts of about 12 s, similar to that of the undisturbed swarm flashing just outside the tent (*Figures 1D and E* and *Figure 2*).

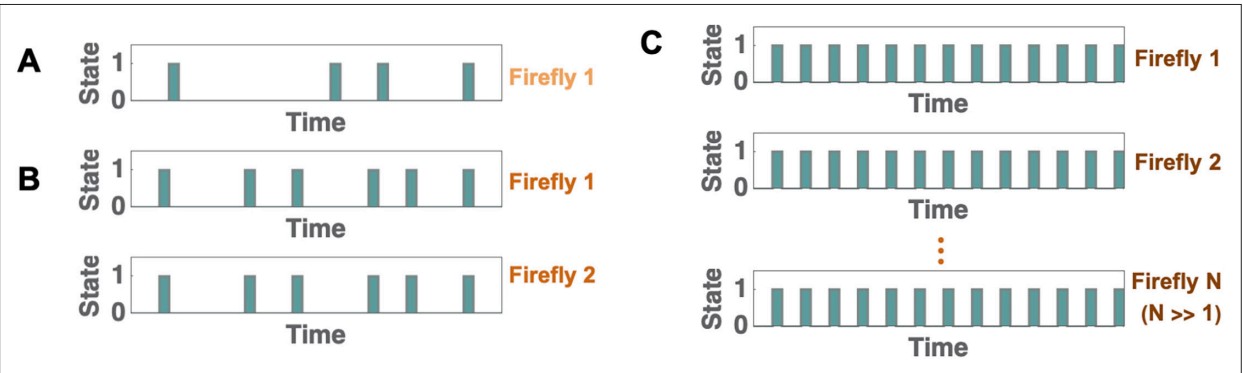

**Figure 2.** Schematic representation of the proposed principle and its theoretical implication: the emergence of periodicity in stochastically flashing *P. carolinus* fireflies. (**A**) A schematic of the flashing pattern of a single isolated firefly. State 0 corresponds to no flashes, and state 1 corresponds to a burst of consecutive flashes. The durations between bursts of single isolated fireflies are highly irregular. (**B**) In a system with more than one firefly, if a non-flashing firefly sees another one flash, it too starts flashing. Thus, for a system with two fireflies, their bursts are synchronized. After each burst, the time to next burst is determined by which firefly flashes first. Thus, on average, the interburst interval is lower, and hence slightly more regular, than that for a single isolated firefly. (**C**) As the number of fireflies increases, the probability increases that at least one of them will flash with an interburst interval near the minimum of the distribution for isolated fireflies. This minimum value is expected to be set by the refractory period of the fireflies, which is expected to be similar for all fireflies. Thus, the overall behavior becomes highly periodic with a period approaching this minimum value.

## Proposed principle of emergent periodicity, its theoretical formulation, and analytic fitting-free predictions

Here we propose the following paradigm, derive its mathematical formulation, and validate its predictions against experimental data: (1) Each time a firefly has finished a burst of flashing, it waits a random time $t$, drawn from a distribution $b(t)$, before flashing again. (2) Upon flashing, a firefly instantly triggers all other fireflies to also flash. (3) After flashing, each firefly resets its internal waiting time to another random $t$.

The distribution $b(t)$ here is the distribution of interburst intervals exhibited by the firefly in a solitary, isolated environment. We denote by $T_b$ the *collective* interburst interval, that is the time between any two successive bursts of flashes produced in the swarm. The probability distribution $P_N(T_b)$ of the interburst interval $T_b$ of a group of $N$ fireflies can be calculated as the probability distribution that one of the $N$ fireflies emits its first flash at time $T_b$ after the last collective burst, while the rest $(N-1)$ fireflies have not flashed until then.

If all fireflies have different IBI distributions such that the interburst interval for the $i^{th}$ firefly in isolation is drawn from the distribution $b_i$, then the probability density for $i^{th}$ firefly flashing first in a group of $N$ fireflies at time $T_b$ is given by

$$P_i(T_b) = b_i(T_b) \prod_{j \neq i} \left[ \int_{T_b}^{\infty} b_j(t)\, dt \right], \tag{1}$$

where the first term on the right is the probability density of the $i^{th}$ firefly flashing at time $T_b$, and the second term is the probability that the remaining fireflies do not flash before time $T_b$. The probability density for any firefly in the group of $N$ fireflies flashing first at time $T_b$ is simply the sum of the probability densities of the individual fireflies flashing first at this time, thus,

$$P_N(T_b) = \sum_{i=1}^{N} b_i(T_b) \prod_{j \neq i} \left[ \int_{T_b}^{\infty} b_j(t)\, dt \right]. \tag{2}$$

As the number of fireflies increases, this distribution converges to a distribution bounded by the minimum and maximum values of the minimum interburst intervals $T_0$ of the individual fireflies. To show this, we first label the minimum interburst interval for $i^{th}$ firefly in isolation by $T_{0,i}$, Thus $b_i(T_b < T_{0,i}) = 0$. Hence, from *Equation 2*, as $N \to \infty$, for $T_b < \min_i(T_{0,i})$, $P_N(T_b) = 0$ because each $b_i(T_b)$ is 0. Also,

$$P_N(T_b) = \sum_{i=1}^{N} b_i(T_b) \prod_{j \neq i} \left[ \int_{T_b}^{\infty} b_j(t)\, dt \right] \leq N \max_i \left[ b_i(T_b) \right] \left\{ \max_j \left[ \int_{T_b}^{\infty} b_j(t)\, dt \right] \right\}^{N-1}. \tag{3}$$

For $T_b > \max_i(T_{0,i})$, as $N \to \infty$, $P(T_b) \to 0$ because the right-most integral is less than 1. Thus, as $N \to \infty$, $P_N(T_b)$ is bounded by the minimum and maximum values that $T_{0,i}$ can take. We expect these minimum values to be set by physiological constraints (the refractory period), and thus be similar for all fireflies. In this case ($T_{0,i} = T_0 \forall i$), the group interburst interval distribution converges to the Dirac Delta function in the large $N$ limit,

$$\lim_{N \to \infty} P_N(T_b) = \delta(T_b - T_0). \tag{4}$$

The theoretical predictions are consistent with the intuitive result that the shortest possible interburst interval is the only one that occurs in large, fully connected, and instantaneously stimulated groups of fireflies. We expect such a threshold minimum time to exist owing to physiological constraints, which prevent the fireflies from flashing continuously forever without pause. Intuitively, as the number of fireflies increases, there is a greater probability that at least one of those fireflies will flash at an interval close to the minimum.

In the following sections, due to the paucity of available data and limited statistical precision in the data available to accurately quantify the IBI distributions for isolated fireflies, we have pooled together the isolated fireflies' data under the assumption that their interburst interval distributions are sufficiently close, so that they can be approximately considered identical ($b_i = b \forall i$). Thus, the interburst interval distribution for $N$ fireflies reduces to

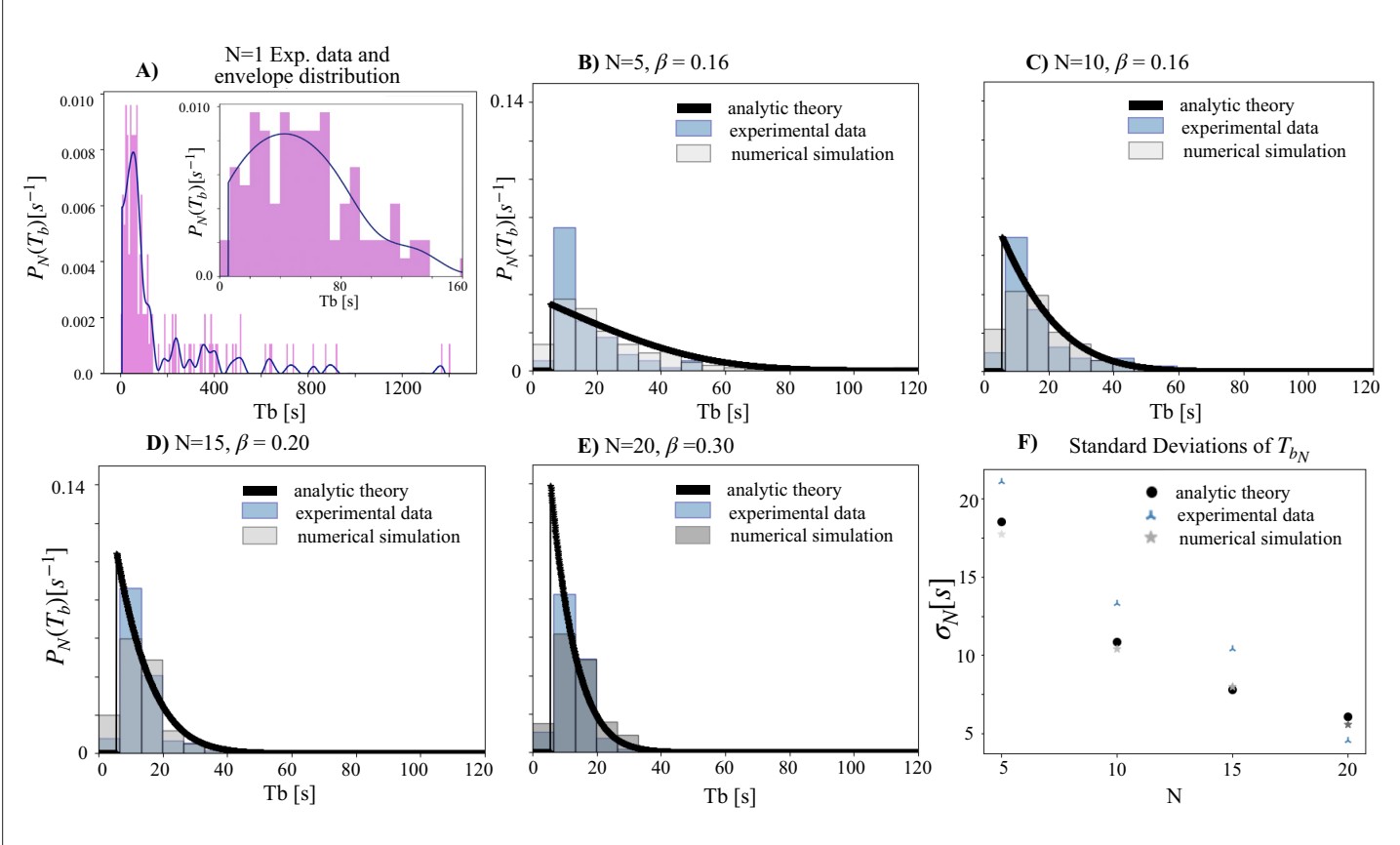

**Figure 3.** Experimental data vis-à-vis results from analytic theory (no fitting parameters) and computational approach (wherein $\beta$ is a fitting parameter as explained in accompanying text). Experimental data for each value of N come from three repetitions of experiments at that density. (**A**) The experimentally measured single firefly interburst distribution (**Figure 1D**, purple, represented here also in purple). The smoothed version of this distribution (blue curve, detailed methods outlined in the Methods Section) is used as an input in analytical theory and, in conjunction with $\beta$ values, in the computational approach. The inset shows the region between 0–160 s within which most firefly values lie. (**B–E**) show the interburst distributions for different numbers of fireflies. Our theoretical framework accurately predicts the sharpening of the interburst distribution as $N$ increases, without the need of fitting parameters. The $\beta$ value atop each figure is fit by minimizing the two-sided Kolmogorov-Smirnov test between the simulation and experimental distributions (see **Figure 7** for a full sensitivity analysis). (**F**) demonstrates that the standard deviation of the interburst interval distribution decreases with $N$ as predicted by analytic theory (no fitting parameter; see theory section) and the computational approach (using the respective value of best-fit $\beta$ shown with the corresponding distribution in **B–E**).

$$P_N(T_b) = N \left[ \int_{T_b}^{\infty} b(t)\, dt \right]^{N-1} b(T_b) \tag{5}$$

Thus we have set up a mathematical framework which takes as its input the experimentally observed interburst distribution and makes specific predictions with no fine-tuning fitting parameters.

Conceptually, in the idealization that at $N \to \infty$ this distribution converges to a Dirac Delta function, which tends to make the flashing patterns perfectly periodic with no variation. However, for a finite number $N$ of fireflies, the distribution peaks at a value greater than $T_0$, and has a specific non-zero width that decreases with increasing $N$ (see section 'Theoretical framework'). These specific predictions are spectacularly borne out by the experimental data. With no fine-tuning fitting parameter, and the experimentally observed single firefly distribution (**Figure 3A**) as the only input to the mathematical framework, we see an excellent match between the $N$-dependent experimentally observed interburst distributions and the corresponding prediction from analytic theory (**Figure 3B–E**). Moreover, the corresponding sharpening of the peak of the distribution (resulting in decreasing noise) with increasing $N$ also quantitatively matches with the trend predicted by theory — see the plot of standard deviation vs. $N$ in panel **Figure 3F**. Through these compelling matches between predictions from

the theory, without fitting parameters, and the experimental observations, we establish the validity of the proposed principle for emergent synchrony and periodicity.

Furthermore, using the analytic framework, the following rigorous results can be generally proved to hold for *any* input single firefly distribution: As the number $N$ of fireflies increases, along with the variance, *all* the moments of the interburst distribution monotonically decrease. In addition, the left-most mode shifts further towards the left with increasing $N$ until it reaches $T_0$. Taken together, what these predictions show is that for any input distribution shape, we are guaranteed to get emergent periodicity and synchrony through the proposed mechanism. We have provided detailed derivations of these predictions in Methods Section (Theoretical Framework).

## Computational approach: Agent-based simulation

In the preceding section, we have articulated a principle of emergent periodicity, its theoretical formulation, and provided concrete fitting-free predictions which are spectacularly borne out by data. Here we attempt to build on the success of theory with an agent-based simulation.

At the outset, we clarify that our attempts at agent-based simulation, which simply tweak extant models, such as Kuramoto or integrate-and-fire (IF), without incorporation of the insights offered by the theory principle, framework, and predictions, fail to reproduce the basic phenomenology observed in data. Instead, we use the insight from theory as an integral building block to reconstruct a computational approach which reduces to the theory in the appropriate limit but leverages the addition of a fitting parameter to incorporate more nuanced considerations. In particular, we now relax the assumption that all fireflies immediately start flashing upon seeing any other one flash, since in practice there could be some time delay or imperfect information transfer, which could be made shorter if the firefly sees additional fireflies flashing too. The rate at which this delay is shortened in proportion to the number of flashing fireflies is given by the behavioral coupling between the fireflies, labeled $\beta$. When $\beta \to \infty$, this limit represents the idealization derived in the theory section: the strongly correlated limit, wherein a single firefly's flashing is sufficient to immediately stimulate all others to also start flashing, while $\beta = 0$ represents completely non-interacting fireflies.

The important distinction between this computational approach and traditional IF models is that as the system becomes more non-interacting (i.e., $\beta$ decreases), the individual behavior becomes more non-oscillatory and sporadic. Thus, incorporating the theoretical framework built up in previous sections is essential to give rise to emergent periodicity despite having non-oscillating individuals.

### Formulation

We propose a simple numerical simulation based on the mechanism previously described. Following previous computational models (*Ramírez Ávila et al., 2011*; *Ramírez Ávila et al., 2003*; *Ramírez Ávila et al., 2019*), we implement a group of $N$ fireflies whose flashing dynamics is governed by charging and discharging processes which represent the time between two bursts and the duration of a burst, respectively. Here, for the sake of simplicity, we simulate bursts of only one flash in length. These processes are determined by both an agent's internal characteristics and its interactions with the group. Specifically, the internal state of firefly $i$ is characterized by variables $V$ and $\epsilon$ whose evolution follows (*Figure 4*):

$$\frac{dV_i(t)}{dt} = \frac{1}{T_{s_i}}\epsilon_i(t) - \frac{1}{T_{d_i}}[1 - \epsilon_i(t)] + \epsilon_i(t)\sum_{j=1}^{N}\beta_{ij}\delta_{ij}[1 - \epsilon_j(t)], \tag{6}$$

which is a standard equation for the IF scheme. Here, $\epsilon_i$ is a binary variable that is 1 when an individual is charging (quiet) and 0 when an individual is discharging (flashing). The state of $\epsilon_i$ changes to 0 when reaching the threshold voltage $V = 1$, and switches back to 1 when the firefly has finished discharging at the threshold ($V = 0$). The time $T_{d_i}$ represents the flash length and is drawn directly from observed data, and the time $T_{s_i}$ represents the end-to-start interflash interval (*Figure 4A*). This value comes directly from the data in the following way: $T_{s_i} = T_{b_i} - T_{d_i}$, where $T_{b_i}$ represents the start-to-start inter-flash interval for firefly $i$, drawn directly from the input distribution envelope in *Figure 3A*. The firefly may be 'pulled' toward flashing sooner if detecting the flashes of neighboring fireflies, which is represented in the framework by the third term (*Figure 4B*). Here $\delta_{ij} \in \{0, 1\}$ represents connectivity between agents and $\beta_{ij}$ is the coupling strength. For simplicity, here we use all-to-all connectivity

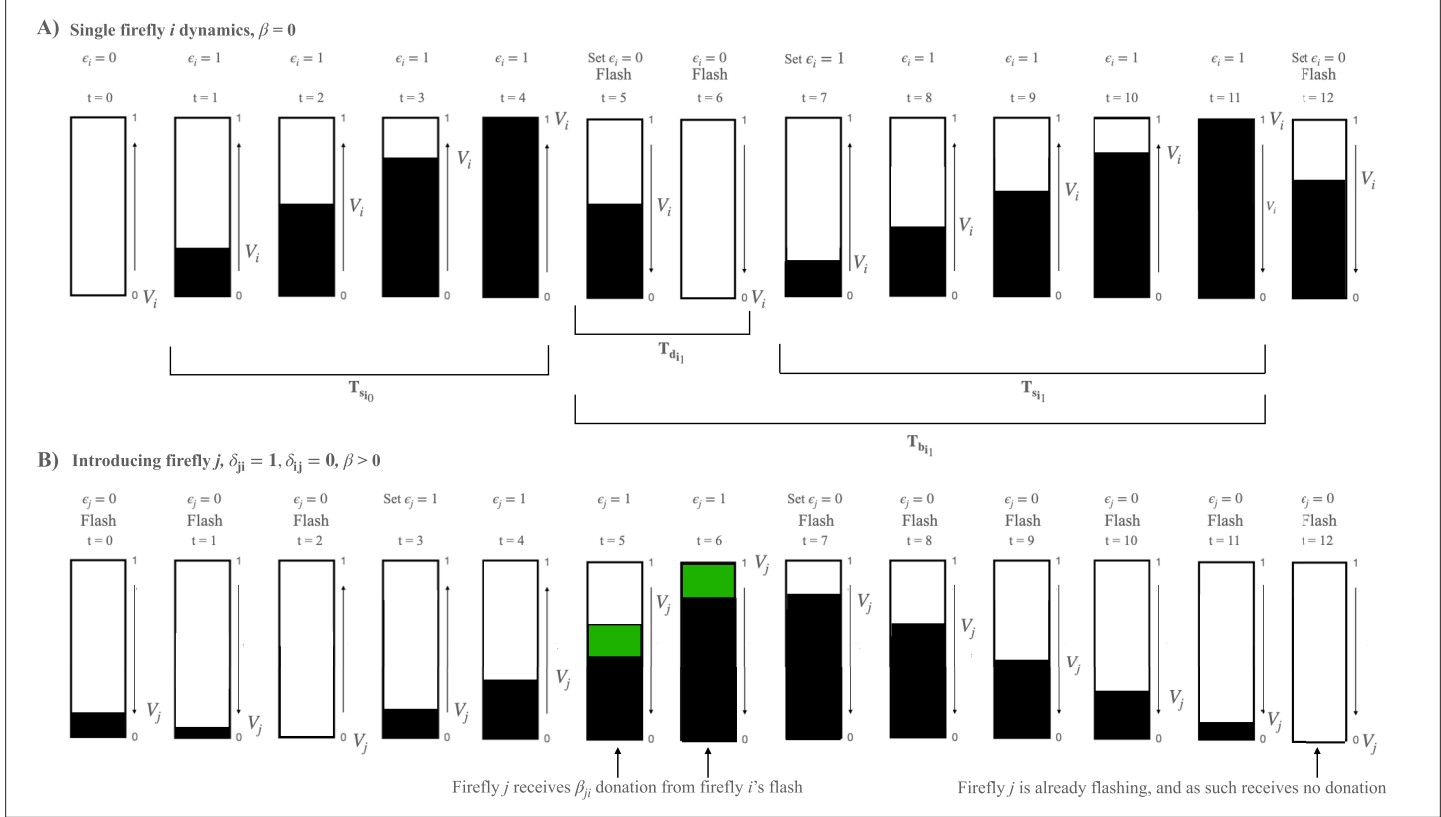

**Figure 4.** A schematic illustrating the computational approach. The dynamics proceed as follows: for each flashing firefly $i$, follow three simple steps at each timestep. (1) Update $\epsilon_i$ according to voltage value. If $V_i == 1$, update $\epsilon_i = 0$; if $V_i$ is 0, update $\epsilon_i = 1$. (2) If $\epsilon_i = 0$, flash. (3) Update their own voltage based on **Equation 6**. (**A**) A single firefly $i$'s dynamics. Dark bars indicate voltage values from 0 to 1. The start-to-start interflash interval $T_{bi}$, end-to-start interflash interval $T_{si}$, and quiet period $T_{di}$, each of which is a random variable for each individual and subject to resampling after each flash event, are indicated below the trace. Flashing state $\epsilon_i$ is indicated above, along with the times at which a flash is being actively emitted by the firefly. (**B**) Schematic of a second firefly $j$, with different parameters, interacting with firefly $i$ via integrate-and-fire $\beta$ donation. For simplicity, we only show a one-way interaction here, where donations occur from firefly $i$ to firefly $j$ and not the reverse. Note the non-linearity in the voltage trace as a flash by firefly $i$ triggers a larger gain in voltage between $t=4$ and $t=5$ and $t=5$ and $t=6$, indicated by the green bars. Firefly $i$'s second flash is ignored by firefly $j$ since it is already flashing ($t=11$, $t=12$).

($\delta_{ij} = 1$, $\forall(i,j)$) and vary the common interaction $\beta_{ij} = \frac{\beta}{N}$. The crucial difference with prior IF implementations is the introduction of stochasticity: $T_{bi}$ is a random variable whose value is drawn from our experimental distributions of interburst intervals (**Figure 3A**), and $T_{di}$ is a random variable whose value is drawn from our previously published data illustrating the distribution of firefly flash lengths, as seen in **Sarfati et al., 2020** (their Figure 7a). Each of these variables resets, for each agent, every time they switch state. In this stochastic IF framework, the variability between flashes is accounted for, while maintaining the overall structure of the IF model.

## Transition to periodicity

This simulation exhibits a transition to group periodicity as interactions between agents are increased. We define the *group* interburst interval as the time between one flash and the next flash produced by *any* other firefly in the swarm. For example, consider the case of $N = 20$ (**Figure 5D**). When $\beta = 0$ each firefly behaves purely individually and interburst intervals tend to aggregate towards small values due to the random unsynchronized flashing of the $N$ fireflies each with a flashing behavior typical of isolated individuals. This remains the case until the coupling strength, $\beta$, becomes large enough that there is enough of collective entrainment to align the flashes of the group. In these regimes, when one firefly flashes, it quickly triggers all others. All agents then reset their charging time at roughly the same moment, and the smallest $T_b$ selected by any individual firefly defines the duration between this flash and the group's next flash. As a consequence, interburst intervals of the collective, $T_b$, shift

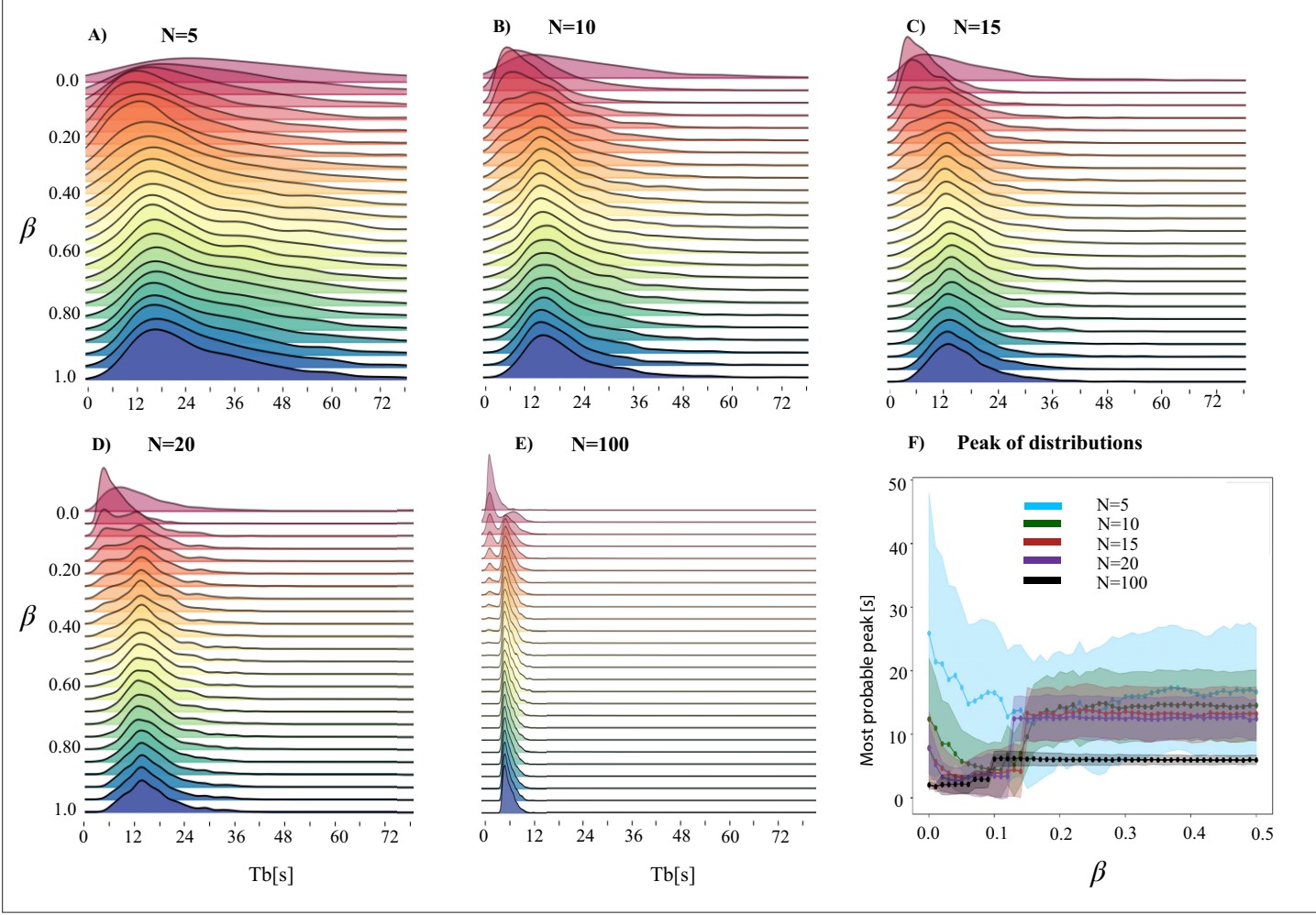

**Figure 5.** Emergence of collective periodicity in large swarms. (**A–E**) Visual demonstration of the emergence of a collective periodicity above $T_{b_0}$ as $\beta$ ranges between 0–1 for each value of N, including (**E**) $N = 100$, a value outside the scope of our experimental observations but that is relevant for the theoretical analysis. The lack of coupling in the first few rows produces noisy and cluttered collective interburst intervals as flashes from any individual are uncorrelated with those from its neighbors. As the coupling constant increases, a consistent interburst interval emerges at the peak of each distribution. (**F**) The relationship between the most probable interburst interval (the distribution peak) as $\beta$ and $N$ vary. The shaded regions represent the standard error of the distributions for each density. For small values of beta, the collective produces noisy distributions where the pulsatile coupling of flashes is not quite enough to pull the starts of bursts into alignment. However, as the coupling constant $\beta$ increases, individual flashes begin to trigger subsequent flashes in neighboring fireflies, causing the quiet periods of the individuals to line up and the emergence of a collective frequency at the fastest interval in each burst cycle. Each higher density simulated causes the peak of the distribution to both shift slightly downwards and become less variant, as it is progressively more likely for one individual in the swarm to drive the collective frequency towards intervals on the short end of the input distribution. A cartoon of this effect is shown in .

to a larger value corresponding to the smallest time between flashes for an individual firefly ($t_{b0}$). This behavior can be seen easily in *Figure 5*, where wide distributions give way to progressively tighter shapes as $\beta$ and $N$ increase. We can quantify this transition by examining the characteristic peaks in the $T_b$ distribution. Peaks with a value below the minimum of the input distribution occur when beta is small, and pulsatile coupling is thus weakly pulling the flashes towards each other. At each value of $N$, however, *Figure 5* shows a sharp transition wherein the beta value becomes high enough to cause enough coupling gain to produce synchronous flashes and the alignment of the start of the next burst. This drives the pace of the flashing to be set by the first flasher, which as $N$ increases becomes more likely to be on the lower end of the input distribution. The high-coupling peak is also naturally sharper at increasing $N$: at larger $N$, the probability that some $T_{b,i}$ approaches the minimum possible $T_b$ is higher, resulting in more regularity the collective flashing pattern (*Figures 3E and 6*).

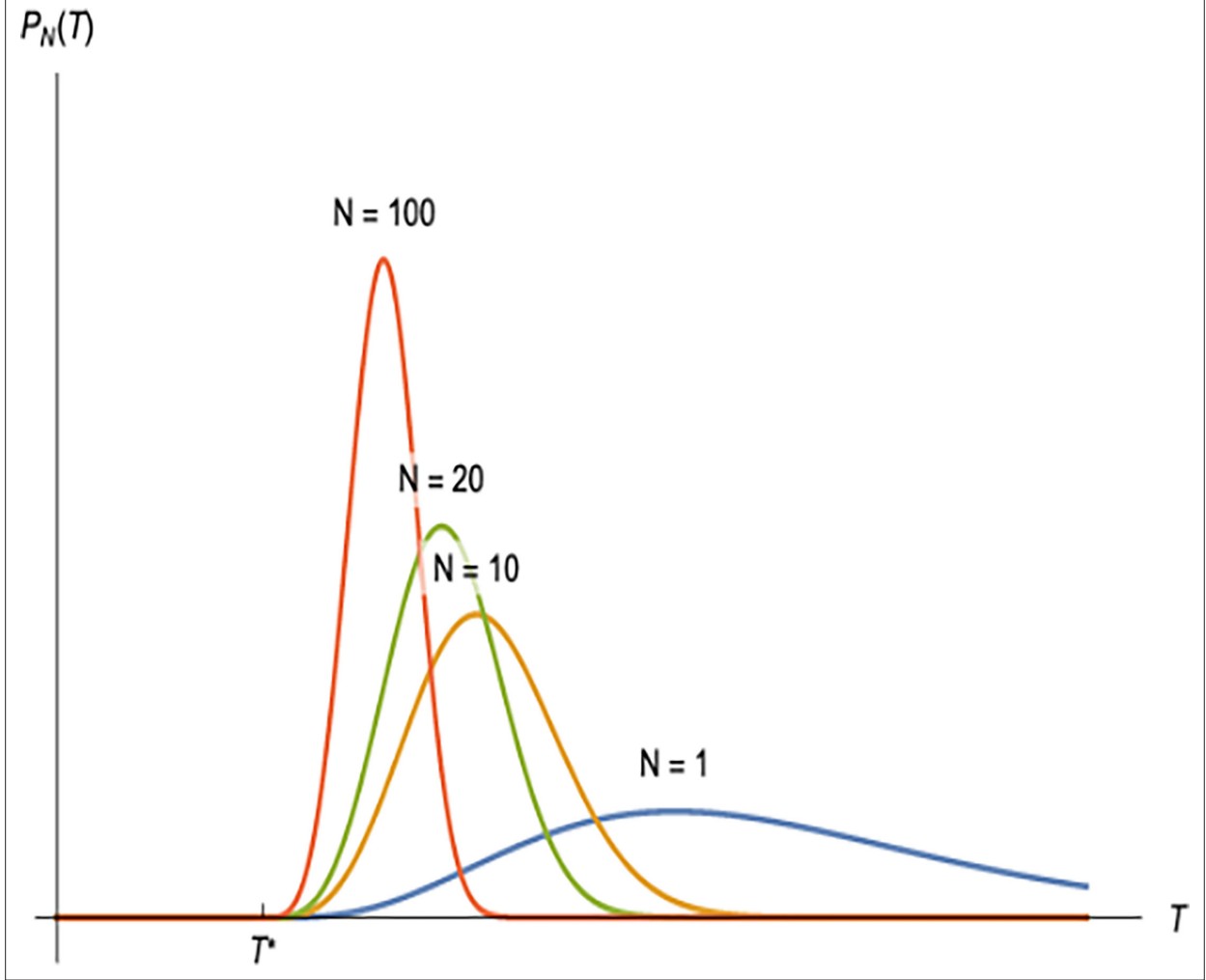

**Figure 6.** Schematic illustration demonstrating the evolution of the collective burst distribution, i.e., the distribution of time intervals between collective bursts, $P_N(T)$, with increasing number of fireflies, $N$. $N = 1$ corresponds to the intrinsic burst distribution of a single firefly, $b$. Evidently, the distribution of time intervals between collective bursts becomes a sharply peaked distribution with maximum probability peaked at a value larger than $T^*$.

As our simulation has a single fitting parameter, namely the coupling strength $\beta$, we conduct a detailed comparison of the simulation and experimental data to infer the most likely value of $\beta$ for the *P. carolinus* system. A systematic parameter sweep over the values of $\beta$ and $N$ provides a set of $T_b$ interval distributions (*Figures 5 and 7*). We statistically compare the distributions generated by simulation with those obtained experimentally at each swarm density and find that the optimal values of $\beta$ to match the empirical distributions cluster around 0.15 when $N \leq 15$ (and holds a higher value when $N = 20$). This also corresponds with the transition point in the location of the mode of the distributions, as can be seen in *Figure 5F*.

## Discussion

In this work, we have proposed a synchronization mechanism that produces emergent periodicity and demonstrated its remarkable quantitative applicability to the synchronous periodic flashing of *P. carolinus* fireflies as observed in natural settings. In other, more commonly studied firefly species such as the *P. frontalis*, individuals are intrinsically oscillatory (*Moiseff and Copeland, 2000*), and thus can be modeled by traditional Kuramoto-like models which do not apply to species like ours. More recently, a new model based on the concept of elliptic bursters has been successful at producing many aspects of *P. carolinus*' collective flashing, notably the intermittent (burst-type) synchrony. Yet,

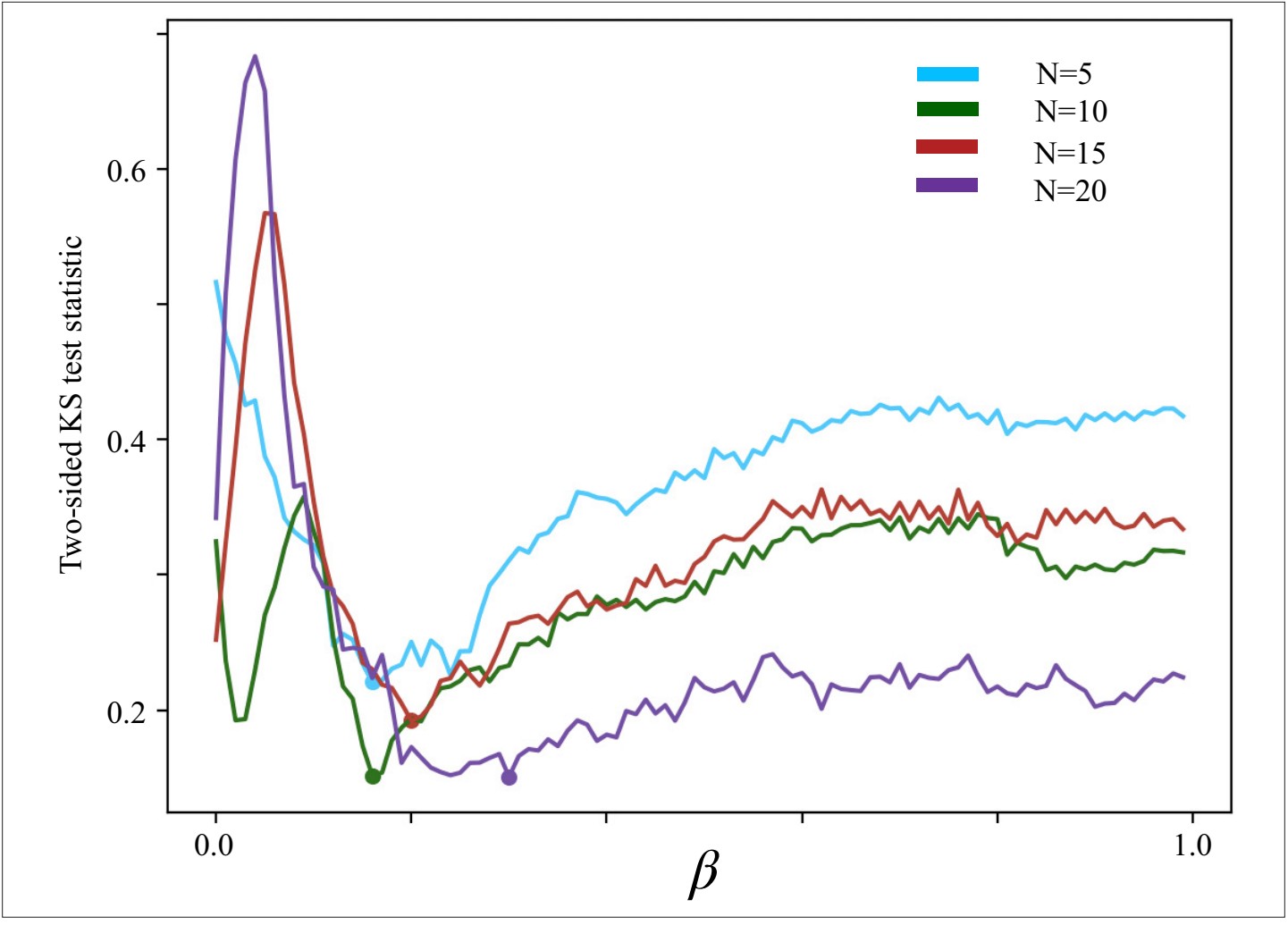

**Figure 7.** Two-sided Kolmogorov–Smirnov test results between the simulation results and experimental results at each $\beta$ and $N$. For the two-sided Kolmogorov–Smirnov test, the null hypothesis states that the two compared distributions can be drawn from the same underlying distribution: effectively, accepting the null hypothesis accepts the statistical probability that the distributions do not differ. All distributions were generated from ten simulations, each of 200000 simulation timesteps/30 min of real time. The best values for each $N = 5, 10, 15, 20$ are $\beta = 0.16, \beta = 0.16, \beta = 0.20, \beta = 0.30$.

this model still assumes intrinsic periodicity between flash bursts (*McCrea et al., 2022*). In systems following our principle, individuals may behave erratically without any periodicity in their behavior, yet when brought together as a collective, their behavioral patterns become highly synchronized and periodic. Moreover, this effect increases with the number of fireflies present through a simple and intuitive behavioral pattern. Using this principle, we successfully predict the qualitative sharpening of the peak of the distribution of interval between flashes by simply using the interval between flashes of isolated individual fireflies and without requiring any fitting parameter. Further, our computational approach quantitatively builds on the predictions of the theory by letting the strength of coupling between fireflies vary and provides added insights.

Specifically, we have shown that the simple theoretical behavioral framework presented in this paper successfully reproduces the experimental distributions of interburst intervals for groups of $N$ fireflies (*Figure 3B and E*). All the input parameters for application of the framework as well as the computational approach come directly from experimental results in *Sarfati et al., 2020* and subsequent field season results from the Great Smoky Mountains: the wide distribution of interburst intervals for single isolated fireflies, the two timescales required by the computational approach of charging time and discharging time are both data-driven from *Sarfati et al., 2020*. The only fitted parameter

for the computational approach is the coupling strength $\beta$, which demonstrates a transition in the dynamics of the system where $\beta > 0.1$ (**Figure 5**).

As shown in **Figure 3**, the chosen values for beta, the additional fitting parameter introduced in the agent-based simulation, are: $\beta = 0.16$, 0.16, 0.20 and 0.30 respectively for $N = 5$, 10, 15, 20. Perhaps it is intriguing that the optimum beta clusters around similar values for $N = 5$, 10, 15, while the optimum beta for $N = 20$ is significantly different. While we do not currently have an explanation for why the fitted parameter values are what they are, we note that the fitting curve is flat, implying that several beta values could possibly achieve a satisfactory fit. Further agent-based simulations could explore these findings more systematically and provide useful insights.

If the number of fireflies increases indefinitely, or if there are visual obstacles in the environment, the assumption that each firefly can practically immediately perceive when another firefly starts flashing will no longer hold. In this case, a finite time delay in perceiving the onset of the flashing could lead to an interburst interval that is greater than what is expected for the ideal case. The resulting interburst interval distribution will consequently be shifted to the right compared to the distribution given by **Equation 5**. While the general ideas underlying the theory framework will continue to hold, the mathematical formulation will need more sophistication to take these subtler effects into account.

Existing mathematical models on synchronous periodic behavior generally consider individuals to be intrinsically oscillatory, which either oscillate periodically in isolation or have their oscillations suppressed at low numbers through a sufficiently strong coupling with the environment. These models generally introduce variability through varying the frequencies of individual oscillators, and synchronization emerges spontaneously once the number or density of these coupled oscillators crosses some specific threshold. Conversely, in our proposed framework, individuals that are intrinsically non-oscillatory make use of the synchronization through coupling with other individuals to produce emergent oscillatory behavior, which becomes more regular as more individuals are added.

Existing mathematical models designed for emergent synchronization of individual oscillators could be extended to account for such behavior by replacing individual oscillators with stochastic sporadically firing individuals. Our framework is simply the simplest version and a starting point for such models. For example, systems of oscillators interacting with Kuramoto-style mean field and limit cycle oscillators such as those used in dynamical quorum sensing models tend to converge on the mean frequency of the heterogeneous group (**Pikovsky et al., 2001**; **De Monte et al., 2007**). However, observations of the *P. carolinus* fireflies show convergence on the fastest frequency in the repertoire of isolated individual fireflies and a synchronization of relaxation periods also seen in some coupled IF units (**Bottani, 1996**) which have been applied to many biological systems such as the synchronization of pacemaker cells in the mammalian heart (**Jongsma et al., 1983**; **Jalife and Michaels, 1989**). Yet the difference lies in the nature of this 'fastest frequency'. In typical coupled IF units, this is the frequency of the individual oscillator with the fastest frequency. But in our system, the individuals fire sporadically, thus there is no specific frequency associated with any individual. Instead, the 'fastest frequency' is an emergent phenomenon in large groups, formed from the collective minimum interburst intervals of the individuals. While individual behavior may appear as extremely complex, collective behavior based on simple and credible behavioral rules converges towards a simple emergent phenomenon as we have demonstrated. This wait-and-start phenomenon might be observable in different biological systems as well.

The mathematical implementation of the proposed paradigm results in an interburst interval distribution that converges towards a unique possible value corresponding to the lower bound of the individual IB distribution, at increasing $N$. That means that in the limit of an infinitely large and entirely connected swarm, the smallest IBI always occurs. This is at odds with two empirical observations: (1) while most of the smallest IBI from an isolated firefly peak at 12 s and more, there are some residual values between 5 s and 12 s; (2) natural swarms comprising thousands of fireflies do not exhibit a 5 s period. We propose some explanation to reconcile these two facts.

First, fireflies are known to produce annex flash patterns, for instance, for alarm, in addition to the primary courtship phase. It is possible that isolated fireflies in a confining volume switch to different behavioral modes that produce atypical flash trains with intervals less than what they would typically do in an unobstructed environment with responding peers. Secondly, it is possible that the swarm buffers against unusual perturbations. More than finite-size effects, the main caveat here is that the swarm is not all-to-all connected, as we showed previously (**Sarfati et al., 2021**). In this case, the

dynamics of the system would depend upon the speed of propagation of information across the swarm.

It is easy to imagine extensions of this work that leverage the spatial positions of individuals in the system using distance- or sight-dependent coupling to modify the adjacency matrix and add further complexity to the system, and this framework makes implementation of this idea ripe for a future endeavor. To provide direct evidence for the underlying mechanistic principles, further experiments are needed. A promising avenue consists of artificially and controllably tuning the interactions within the group, for example, artificial flash entrainment with an LED should be able to decrease the interburst interval.

## Materials and methods

### Experimental data

The individual and collective flashing of *P. carolinus* fireflies was recorded during 10 nights of field experiments in June 2020 in Great Smoky Mountains National Park (Tennessee, USA). The experimental protocol had been developed and implemented the previous year (*Sarfati et al., 2020*). In the natural swarm with hundreds to thousands of interacting fireflies, collective flashing consists of synchronous flashes every $T_\text{f} \simeq 0.5\text{s}$, during periodic bursts $T_\text{b} \simeq 12\text{s}$ (*Figure 1C*). However, it has been observed previously that individual fireflies in visual isolation do not exhibit burst periodicity. To characterize the onset of burst flashing, we performed experiments in a controlled environment. Fireflies were gently collected using insect nets, then placed individually in small plastic boxes, where species and sex were verified. Males were subsequently introduced into a secluded cuboid tent (approximately $1.5 \times 2 \times 1.5\text{m}^3$) made of breathable black fabric and covered by a black plastic tarp to ensure optimal visual isolation from fireflies on the outside. A GoPro Fusion 360° camera placed inside the tent recorded the entire volume at 30 or 60 frames-per-second (fps). Flashes were detected in video processing by intensity thresholding. Bursts were identified as (temporal) connected components of flashes less than 2 s apart. Interburst intervals $\tau_\text{b}$ were calculated as the duration between the first times of successive bursts. Tent experiments allow us to observe the collective behavior of a small and known number of fireflies in interaction, while providing enough space for them to fly, hence reducing experimental artifacts from excessive confinement. We observed the flashing behavior of both individual fireflies in isolation and groups of 5, 10, 15, and 20 fireflies. We observed 10 individual fireflies alone in the tent, over durations between 5 min and 85 min. We observed that although these fireflies produced flash trains at a frequency of about 2 Hz, the delay between successive trains was apparently randomly distributed, from a few seconds to tens of minutes. Then, we carried out three sets of experiments with 5, 10, 15, and 20 fireflies, using the segments between 9 min and 15 min. As previously reported, collective burst flashing only appears at about 15 fireflies.

### Experimental data correction

After the paper's acceptance, a small subset of data points was updated for the reasons described in the correction notice (*Sarfati et al., 2025*). We repeated all analyses and confirmed that the findings are unaffected. Both the original and corrected datasets are publicly available.

### Theoretical framework

#### Behavior of moments and variance

For the following sections, we assume that individual isolated fireflies have identical interburst interval distributions. We show that as the number of fireflies ($N$) increases, the variance and all the moments of the interburst interval distribution decrease and the distribution eventually converges to a Dirac Delta function. From *Equation 5*, the $m^\text{th}$ moment for $N$ fireflies is

$$\langle T_N^m \rangle = N \int_0^\infty \left[ \int_t^\infty b(t')\, dt' \right]^{N-1} t^m b(t)\, dt. \tag{7}$$

Let the function $\gamma$ be defined as

$$\gamma(t) = \int_t^\infty b(t')\, dt', \tag{8}$$

thus,

$$\langle T_N^m \rangle = -N \int_{t=0}^\infty \gamma^{N-1}(t) t^m \, d(\gamma(t))$$
$$= -\gamma^N(t) t^m \Big|_0^\infty + m \int_0^\infty \gamma^N(t) t^{m-1} \, dt. \tag{9}$$

We expect the distribution of inter-burst intervals to terminate at some large value and not go on to infinity (at most, they are limited by the finite lifespan of the fireflies), thus,

$$\langle T_N^m \rangle = m \int_0^\infty \gamma^N(t) \, t^{m-1} \, dt. \tag{10}$$

Now, at any given value of $t$, $\gamma^N(t) \leq \gamma^{N-1}(t)$. This inequality is strict whenever $0 < \gamma(t) < 1$. Such a region exists unless $b(t)$ is a Dirac Delta function. If $b(t)$ is a Dirac Delta function, then $P_N(T_b) = b(T_b)$. Otherwise,

$$\int_0^\infty \gamma^N(t) \, t^{m-1} \, dt < \int_0^\infty \gamma^{N-1}(t) \, t^{m-1} \, dt, \tag{11}$$

$$\Rightarrow \langle T_N^m \rangle < \left\langle T_{N-1}^m \right\rangle. \tag{12}$$

Thus, all moments strictly decrease as $N$ increases. From *Equation 10*, the variance for $N$ fireflies is

$$V_N = 2 \int_0^\infty \gamma^N(t) \, t \, dt - \left[ \int_0^\infty \gamma^N(t) \, dt \right]^2 \tag{13}$$

Writing the second term initially as a multiple integral over the entire $t, t' > 0$ plane,

$$\left[ \int_0^\infty \gamma^N(t) \, dt \right]^2 = \iint \gamma^N(t) \gamma^N(t') \, dt \, dt' = 2 \iint_{t>t'} \gamma^N(t) \gamma^N(t') \, dt \, dt'. \tag{14}$$

In the preceding step, we have used the symmetry of the integrand under $t'$. The second term of *Equation 13* can be similarly written down:

$$2 \int_0^\infty \gamma^N(t) \, t \, dt = 2 \iint_{t>t'} \gamma^N(t) \, dt \, dt'. \tag{15}$$

Combining,

$$V_N = 2 \iint_{t>t'} \gamma^N(t)(1 - \gamma^N(t')) dt \, dt'. \tag{16}$$

Thus,

$$V_{N+1} - V_N = 2 \iint_{t>t'} \left[ \gamma^{N+1}(t)(1 - \gamma^{N+1}(t')) - \gamma^N(t)(1 - \gamma^N(t')) \right] dt \, dt'. \tag{17}$$

The two $\gamma$ functions in the above integrand satisfy: $0 \leq \gamma(t) \leq \gamma(t') \leq 1$, using the properties of the cumulant function. Thus,

$$\gamma(t')\gamma(t) \leq \gamma(t),$$
$$\Rightarrow \; 1 - \gamma(t) \geq 1 - \gamma(t)\gamma(t') \geq \gamma^N(t') \left[ 1 - \gamma(t)\gamma(t') \right],$$
$$\Rightarrow \; \gamma^N(t) \left[ 1 - \gamma(t) \right] \geq \gamma^N(t)\gamma^N(t') \left[ 1 - \gamma(t)\gamma(t') \right],$$
$$\Rightarrow \; \gamma^N(t) - \gamma^{N+1}(t) \geq \gamma^N(t)\gamma^N(t') - \gamma^{N+1}(t)\gamma^{N+1}(t'). \tag{18}$$

Rearranged, this tells us that the integrand in *Equation 17* is non-positive (i.e., $\leq 0$) everywhere. Thus, we have proved that $V_{N+1N}$. In other words, the variance of the flashing distribution monotonically decreases with increasing number of fireflies.

Further, as $N \to \infty$, $\gamma^N(t) \to 0$ for all $t$ above $T_0$ (which is the maximum value of $t$ below which $b(t)$ is 0). For values of $t$ below $T_0$, $\gamma^N(t) = 1$ irrespective of $N$. Thus, from *Equation 10*,

$$\lim_{N \to \infty} \langle T_N^m \rangle = m \int_0^{T_0} t^{m-1} \, dt = T_0^m, \tag{19}$$

which represents moments of the Dirac Delta function $P_{N \to \infty}(T) = \delta(T - T_0)$ . Thus, as the number of fireflies tends to infinity, the distribution of interburst intervals tends to a Dirac Delta function peaked at $T_0$.

## Behavior of mode

For a single firefly interburst interval distribution $b(t)$ that is continuous for $t_0$ and differentiable for $t > T_0$ (where $T_0$ is the maximum value of $t$ below which $b(t)$ is 0), we show that the left-most mode shifts to the left as the number of fireflies ($N$) increases, unless it reaches $T_0$, in which case it stays at $T_0$ on increasing $N$.

The mode would be the local maximum of distribution $P_N$. Differentiating *Equation 5*,

$$P_N'(t) = N\gamma^{N-2}(t)[\gamma(t)b'(t) - (N-1)b^2(t)]. \tag{20}$$

Let the left-most mode of $P_N$ be located at $t = t_N^*$. If $t_N^* = T_0$, we have

$$\lim_{t \to T_0^+} \gamma(t)b'(t) - (N-1)b^2(t) < 0. \tag{21}$$

Now, on increasing the number of fireflies by 1, we still have

$$\lim_{t \to T_0^+} \gamma(t)b'(t) - Nb^2(t) < 0 \Rightarrow \lim_{t \to T_0^+} P_{N+1}'(t) < 0. \tag{22}$$

Thus, the mode stays at $T_0$ . On the other hand, if $t_N^* > T_0$ , we have

$$\gamma(t_N^*)b'(t_N^*) - (N-1)b^2(t_N^*) = 0. \tag{23}$$

Now, on increasing the number of fireflies by 1, we get

$$\gamma(t_N^*)b'(t_N^*) - Nb^2(t_N^*) < 0 \Rightarrow P_{N+1}'(t_N^*) < 0. \tag{24}$$

Thus, $P_{N+1}$ increases toward the left of $t_N^*$, i.e., $T_0 \leq t_{N+1}^* < t_N^*$ . Thus, the left-most mode shifts to the left as the number of fireflies ($N$) increases, unless it reaches $T_0$ , in which case it stays at $T_0$.

## Numerical demonstration

We use numerical calculations to demonstrate how synchronized periodicity arises in an arbitrary system which follows the extreme-value statistics used in our theory. Here, we take an arbitrary probability distribution (given by $N = 1$ label in *Figure 6*) and plot the distribution of the minimum of $N$ samples obtained from the $N = 1$ distribution. The distributions for arbitrary $N$ are described by *Equation 5* as derived previously. As $N$ increases, these distributions become sharply peaked with maximum probability peaked at a value larger than the minimum of the $N = 1$ distribution. For a system in which these quantities represent the interval between events, for large $N$, those events would become highly periodic as the width of the distribution narrows.

## Agent-based simulations implementation details

### Preparing input for the simulations

The input distribution for the simulations' inter-burst interval $T_b$ is sampled directly from envelope distributions that encapsulate observations of one firefly's inter-burst interval. These envelope distributions were generated using an interpolating $\beta$-spline between bin centers of the histogram of the

distribution, normalized so that the area underneath the envelope sums to 1. The protocol for generating this envelope distribution is as follows:

1. Read and clean the data.
   a. Read the experimental observations of individual firefly $T_b$ from an input file and save into a list called tbs.
   b. Remove from tbs all values below 2.0 s: these were deemed to be 'interflash' values and should not be included.
   c. Extract minimum value $T_{b_{min}}(= 5.672s)$ and maximum value $T_{b_{max}}(= 1403.385s)$ from tbs.
2. Generate the envelope.
   a. Make a density histogram of the data tbs such that $H(x)=y$, defined between $Tb_{min}$ and $Tb_{max}$. Here we used a bin width of 27.954 s (dividing the interval uniformly into 50 bins) to smooth over the bumpy peaks of the input distribution.
   b. From the bins and heights of $H(x)$, compute the coefficients of a cubic spline function (**Press et al., 1992**). This produces a function $B(x)$ that will fit to the 'envelope shape' of the distribution when applied to a particular domain of $xs$.
   c. On the domain $x = [Tb_{min}, Tb_{max}]$ with 0.1 s increments, let $H'(x)=B(x)$. This produces a fine-grained version of the envelope from which samples can be generated later.
   d. $Tb_{min}$ Pad $H'(x)$ with 0 s from $[0, Tb_{min}]$ with bin size 0.1. Call the resulting piecewise function $H''(x)$, where $H''(x)=0$ if $0 < x < Tb_{min}$ and $H'(x)$ otherwise. Thus,

$$H''(x) = \left\{ \begin{array}{ll} 0, & \text{if } 0 < x < Tb_{min} \\ H'(x), & \text{if } Tb_{min} \leq x \leq Tb_{max} \end{array} \right\}. \tag{25}$$

   e. For $x$ in $[0, Tb_{max}]$ with 0.1 s increments: write $(x, H''(x))$ to a new file.
3. Draw from the envelope
   a. Read $(x, H''(x))$ pairs from new file
   b. Let N = number of input values to choose
   c. Randomly sample N values with replacement from distribution, call it tbs2
   d. Instantiate each of N agents with one value from tbs2 and run simulation

## Simulation parameters

All experiments carried out with this agent-based framework were conducted via simulation. The simulation outputs a time series of flashes and their positions. For each set of parameters, we ran simulations for thirty trials of 200,000 timesteps each. Parameters can be varied run-by-run via command-line arguments, which made a grid search parameter sweep over coupling strength $\beta$ and number of fireflies $N$ easily parallelizable. All other values required for the synchronization dynamics are instantiated from experimental observations as explained in the main text.

## Acknowledgements

OP acknowledges internal funds from the BioFrontiers Institute, and a seed grant from the Interdisciplinary Research Theme on Autonomous Systems and the University of Colorado Boulder. SI-B and KJ thank the Purdue Research Foundation, the Showalter Trust, and the Ross-Lynn Fellowship award for financial support.

## Additional information

### Funding

| Funder | Grant reference number | Author |
| --- | --- | --- |
| Purdue University West Lafayette | Ross-Lynn Fellowship | Kunaal Joshi Srividya Iyer-Biswas |
| Showalter Trust | | Kunaal Joshi Srividya Iyer-Biswas |

| Funder | Grant reference number | Author |
|---|---|---|
| Purdue Research Foundation | | Kunaal Joshi Srividya Iyer-Biswas |
| BioFrontiers Institute | | Orit Peleg |
| Interdisciplinary Research Theme on Autonomous Systems and the University of Colorado Boulder | | Orit Peleg |

The funders had no role in study design, data collection and interpretation, or the decision to submit the work for publication.

## Author contributions

Raphael Sarfati, Collected the data; analyzed the data; Designed the computational framework with insights from KJ; Discussed the results; Wrote the paper; Kunaal Joshi, Conceptualized the theory framework and proposed the principle; Performed analytic calculations with contributions from RS; Discussed the results; Wrote the paper; Owen Martin, Analyzed the data; Designed the computational framework with insights from KJ; Performed simulations; Discussed the results; Wrote the paper; Julie C Hayes, Collected the data; Discussed the results; Wrote the paper; Srividya Iyer-Biswas, Conceptualized the theory framework and proposed the principle; Performed analytic calculations with contributions from RS; Discussed the results; Wrote the paper; Supervised research; Orit Peleg, Collected the data; Designed the computational framework with insights from KJ; Discussed the results; Wrote the paper; Supervised research

## Author ORCIDs

Raphael Sarfati ⬡ https://orcid.org/0000-0003-4944-0632
Kunaal Joshi ⬡ https://orcid.org/0000-0002-8001-1230
Srividya Iyer-Biswas ⬡ https://orcid.org/0000-0002-1587-6780
Orit Peleg ⬡ https://orcid.org/0000-0001-9481-7967

## Decision letter and Author response

Decision letter https://doi.org/10.7554/eLife.78908.sa1
Author response https://doi.org/10.7554/eLife.78908.sa2

## Additional files

### Supplementary files

MDAR checklist

### Data availability

The data and code that support the findings of this study are openly available in the EmergentPeriodicity GitHub repository found at https://github.com/peleg-lab/EmergentPeriodicity (copy archived at *Martin, 2025*).

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
