## [Editor Report]

This important study provides a quantitative characterization and understanding of firing collective patterns in P. Carolinus fireflies. The work significantly contributes to fill the gap between observations and mechanistic models, with convincing experimental evidence and solid theoretical modeling. This work will be of interest to readers curious about collective behavior, biological rhythms, and models of synchronized oscillations.

---

## [Decision Letter]

**Decision letter after peer review:**

Thank you for submitting your article "Emergent periodicity in the collective synchronous flashing of fireflies" for consideration by *eLife*. Your article has been reviewed by 2 peer reviewers, and the evaluation has been overseen by a Reviewing Editor and Aleksandra Walczak as the Senior Editor. The following individual involved in review of your submission has agreed to reveal their identity: Steven Strogatz (Reviewer #1).

Both referees have expressed positive comments on the experimental findings and on the modeling part. They raised, however, several concerns/issues that ALL need to be addressed in a revised version. Please look at the referees' reports to understand what their main criticisms are and to read their detailed comments. Please note that Referee 1 also provided an annotated pdf with all comments marked directly in the text. You should be able to download the pdf from the website, if not please contact the editorial office for help.

As you will see, there are a few general points that we consider essential revisions for further review:

1) Some hypothesis, and in particular the one that all individuals have the same inter-burst interval distribution should be tested/justified/discussed.

2) Comparison between the models and the data must be improved, in particular through a quantification of the differences between distributions and sensitivity analysis of the numerical results.

3) More discussion of the modeling in connection to past theoretical results and existing literature is necessary to better contextualize the present work and assess its originality.

*Reviewer #1 (Recommendations for the authors):*

The authors propose a simple model for flash dynamics in a certain species of firefly known as P. Carolinus. Remarkably, individual males of this species flash haphazardly, with no particular rhythm, yet in sufficiently large groups, they somehow manage to flash in rhythmic unison. The authors show that their model can account for this phenomenon - with no adjustable parameters - and they test their model quantitatively in idealized experiments on groups of up to 20 fireflies confined to a small, darkened, cuboid tent of dimensions 1.5 x 2 x 1.5 cubic meters.

Strengths:

The authors' model is unusual in that the individual fireflies are not assumed to be intrinsically rhythmic. By contrast, in most previous work on firefly synchronization the individual fireflies were modeled as "oscillators." That convenient assumption allowed a large body of theory from nonlinear dynamics to be imported. However, for the particular species of firefly being studied here, the authors show the individual males do not flash rhythmically. The authors provide a framework for dealing with this novel case.

There are actually two parts to the framework. In the first (extremely stylized) model, the authors assume that after it flashes, each firefly waits a random time before it can flash again. All fireflies choose this random waiting time from the same probability distribution; in that sense, the fireflies are assumed to be statistically identical. Furthermore, the authors assume that when one firefly flashes, it triggers all the rest to flash instantaneously. A weakness of this extreme idealization is that synchrony is thereby built in automatically by assumption, rather than explained as a consequence of the model. But a virtue of this extreme idealization is that it correctly predicts how the variance in the interburst intervals depends on the number of fireflies in the group. In this way, the authors neatly explain a property of the emergent period that they observe in their tent experiments. Most notably, they do this without any adjustable parameters. The result follows as a consequence of their model and their measurements of individual firefly behavior. This is a beautiful instance of using individual behavior to predict collective behavior with the help of some simple, additional assumptions.

The second part of the new framework builds on existing research on a class of oscillators known as "integrate and fire" oscillators. The new wrinkle is that the authors introduce a stochastic term in the equation (a random amount of time for the oscillator to charge up) in order to capture the erratic nature of the interburst interval for individual males of this firefly species. The virtue of this more complex model is that it allows the authors to predict a transition to group periodicity as the interaction strength between the fireflies is increased. It also allows the authors to relax the earlier (unrealistic) assumption of instantaneous triggering of all other fireflies whenever one firefly flashes.

Weaknesses:

The work presented here is an excellent start at understanding the collective behavior of this particular species of firefly. However, the model does not apply to other species in which individual males are intrinsically rhythmic. So the model is less general than it may appear at first.

The modeling framework is also developed under the very stylized conditions of experiments conducted in a small tent. While that is a natural place to begin, future work should consider the conditions that fireflies encounter in the wild. Swarms that are spread out in space would require a model with a more complicated structure, perhaps with network connectivity and coupling strengths that both change in time as fireflies move around. This is not so much a weakness of the present work as a call to arms for future research.

Overall, the paper does an excellent job of supporting its conclusions with elegant arguments and experiments.

This assumption that all individuals have the same IBI distribution could be directly tested. Has this been done? If not, why not? e.g. Are there difficulties with letting one firefly flash long enough to collect sufficient data to fill out the distribution?

The derivation given in 6.2.1 is clearer than the approach taken here, which unnecessarily introduces Q, q, and c and then never uses them again.

*Reviewer #2 (Recommendations for the authors):*

Synchronous flashing of fireflies is a textbook example of collective behaviour, although much more theoretical work has been devoted to explaining it than field observations. The work proposed in this manuscript, along with previous results by the same group, are significantly contributing to fill the gap between observations and mechanistic models. In these models, single-insect dynamics is described along with the interaction mechanisms.

In a clever experiment where fireflies are screened from the rest of the population, so that the number of interacting individuals can be manipulated, the authors were able to characterize both single-fireflies firing patterns, and the emergence of collective-level flashes when their number is progressively increased. These observations show that collective-level firing is more regular and more frequent that the average individual-level firing, a feature that can be explained by globally coupled populations of dynamical systems, where firing is followed by a refractory period. The authors show at first that the increase in coherence (standard deviation of the inter-spike interval of the synchronous flashes) as a function of the number N of fireflies is quantitatively explained knowing just the inter-burst interval of collective flashing at high density - which is largely determined by the duration of the refractory period. Then, by using an integrate-and fire model that explicitly accounts for coupling, they study the dependence of the onset of collective oscillations on coupling, and estimate what coupling strength best explains the observations at different N.

I found that the experiment is really nice and has the potential to advance a lot the mechanistic understanding of this collective behaviour. Also, the mathematical model reproduces the main observations, even though density-dependent onset of synchronization, reduction of the oscillation frequency with increasing density, and finite-size scaling are general properties that are observed in populations of globally coupled dynamical systems other than the one proposed here.

However, the way models reflect experimental observations and how they compare with them and with one another are in my opinion insufficiently characterized and discussed.

I raise two main points:

1. The biological relevance of certain hypotheses is insufficiently discussed. This is important because if the observed behaviour is a universal one, alternative models may explain it as well.

2. Comparison between the models and the data could be improved, in particular through quantification of the differences between distributions and sensitivity analysis of the numerical results.

I detail below the main points where I think the manuscript could be improved. In general, I think that toning it down a bit would not be out of place. There were also a number of edits that I could signal if I receive a version of the paper with line numbers.

A. The assumption that single-firefly spikes obey the same distribution (there is no individual variation in the frequency, or even of the composing number of bursts, of the flash) does not seem to have been verified on the data, that are instead pulled together in one single distribution (Figure 1D). Moreover, the main feature of such distribution is that it has a minimum at 12 secs (discarding the faster bursts that are not considered in the model) and that it is sufficiently skewed so that it takes a minimal coupling for collective synchrony to emerge. I think that the agreement between the distributions for different N would be more meaningfully discussed having previous work as a reference, whereas now this is relegated to the discussion, so that it is unclear how much of the theoretical results are novel and/or unexpected. Quantification of the distance between distributions would also be interesting: it looks like the two models (analytical and simulations) disagree more among themselves than with the data.

B. If I understand correctly, simulations are introduced as a way to get a dependence on the intensity of the coupling (\β). There are several issues here. First, I do not see how the coupling constant could change in the present experimental setup, where all fireflies presumably see each other (different from when there is vegetation). Second, looking at Figure 3, the critical coupling strength appears to depend very weakly from N, and it is not clear how the 'detailed comparison' that leads to the fit is realized (in fact, the fitted \betas look larger that those at which the transition occurs in Figure 3A). I think a sensitivity analysis is needed in order to understand how do results change when \β is changed, and also what is the effect of the natural Tb distribution (Figure 2 F). Results of the simulations might be clearer if instead of using the envelope of the experimental results, the authors tried to fit it to a standard distribution (ex. Poisson) so that it can be regularized. This should allow to trace with higher resolution the boundary between asynchronous and synchronous firing.

C. More care should be put in explaining what are the initial conditions hypothesized for the different models. For instance, the results of paragraph 3 are understandable if all fireflies are initialized just after firing, something that is only learnt at the end of the paragraph. I also wonder whether initial conditions may be involved with T_bs in the low-coupling region of Figure 3A not being uniformly distributed, as I would have expected for a desynchronized population.

D. I found that equations were hard to understand either because one of the variables was not precisely (or at all) defined, or because some information was missing:

Equation 1: q is not defined

Equation 2: explain what it means: the prob. that others have not flashed times that one flashes. Also, say explicitly what is the 'corresponding PDF.

Equation 3: the equation for \epsilon(t) to which this is coupled is missing

Why introduce \β_{i,j} and T_bi if they are then taken independent of the indexes?

Definitions of collective and group burst interval should be provided.

It would be clearer if t_b0 was defined in the first paragraph of the results, so as to clarify as well its relation with T_b.

Define T^i_b in the caption of Figure 3 (they are defined later than the figure is first discussed).

The definition of 'the vertical axis label' (maybe find a word for that…) is pretty cumbersome. I could imagine that other definitions would allow the lines in Figure 3 E to converge to the same line for large betas, which would make more sense, considering that in the strong coupling limit I see no reason why the collective spiking should not be the same for different N (the analytical model could help here).

E. I think that the author's reading of the two 'dynamical quorum sensing' papers they cite is incorrect: De Monte et al. was not about the Kuramoto model, but the same limit cycle oscillators as in Strogatz; Taylor et al. considers excitable systems, potentially closer to noisy integrate-and-fire, at least in that they do not have self-sustained oscillations. Both papers show that oscillations appear above a certain density threshold, and that the frequency of oscillations increases with density, as found in this work. A more accurate link to previous publications in the field of synchronization theory, including the models by Kurths and colleagues for fireflies, would be useful both in the introduction and in the discussion, and would help the reader to position this work and appreciate its original contributions.

F. The authors say that part of the data is unpublished. I guess they mean that the whole data set will be published with this manuscript. I think the formulation is ambiguous.

[Editors' note: further revisions were suggested prior to acceptance, as described below.]

Thank you for resubmitting your work entitled "Emergent periodicity in the collective synchronous flashing of fireflies" for further consideration by *eLife*. Your revised article has been evaluated by Aleksandra Walczak (Senior Editor) and a Reviewing Editor.

The manuscript has been improved significantly, and both referees praise its enhanced clarity and presentation. There are, however, a few remaining issues that need to be addressed, as outlined below:

Reviewer 2 suggested further comments and more discussion to be added in connection to (i) comparison with other models; (ii) agreement between modeling and observations. All the points raised in the report (see below) should be addressable in a relatively short time, and would further improve the potential impact and outreach of the manuscript.

*Reviewer #2 (Recommendations for the authors):*

The authors have significantly improved the manuscript, where assumptions and analytical and numerical results are now presented more clearly.

I still have some comments, more of less specific, that I list below, starting with the conceptual ones.

1. Citation of previous work on dynamical quorum sensing (lines 51 and 52) I think misses two important points: first these works (and others following them) deal with the appearance of collective oscillations at high density (therefore, the same general problem addressed here); second, Taylor et al. studied also a transition where the oscillators involved did not oscillate at low density, whereas above a density threshold, they display coherent collective oscillations whose period decreases with density – similar to what observed here. I do not think this takes anything away from the originality of this work, which refers to a different system, and models it with different equations, but the parallelism between integrate-and-fire dynamics with quenched noise and excitable dynamics in the presence of noise should in my opinion not be overlooked.

2. As the authors stress in lines 105 and 132, the analytical model shows that all that really matters in this phenomenon is the fastest frequency of the system. This could be used as an argument to say that the actual frequency distribution of individual fireflies is not all that important, as long as their fastest frequency is comparable. The assumption that they are identical would then sound less radical. Ideally, one could use the numerical simulations to check this, as well as the fact that the phenomenon does not break down when the shortest individual interburst interval Tb_min is narrowly distributed (which could also explain why having a few individuals who can flash at a higher frequency does not affect the outcome).

3. I still feel that the agreement between the model and observations is a bit overstated (line 120). At least, I think the authors may stress that whereas the model predicts that the frequency of the 7-14 minutes oscillations should increase a lot with N, this is not observed in the data. Maybe this mismatch would be reduced if inter-individual variability was added.

4. In paragraph 4.2, I found it unclear why the authors find it unsurprising that different experiments would correspond to different betas. I think that this point should be discussed, as β and N appear in combination in determining the interaction strength. Otherwise, they could try to fit all distributions with the same β, which would be more natural for me. I guess that the fits would be anyway good to the eye, though quantitatively suboptimal (which could be quantified with the distance introduced).

---

## [Author Response]

Essential revision:1) Some hypothesis, and in particular the one that all individuals have the same inter-burst interval distribution should be tested/justified/discussed.

a) We have generalized the theory to directly address this point by relaxing the assumption of an identical inter-burst interval for all individuals. In short: the main insights continue to hold and we discuss the nuances in the text.

(b) Experimentally, the hypothesis that all single fireflies isolated from the group exhibit the same interburst interval (IBI) distribution could not be rigorously tested. The main reason is practical: in order to compare IBI distributions across individuals, we would need to collect a large number of fireflies and track them for long durations, which was not realistic given our experimental setup and the short window of firefly emergence. In addition, external environmental factors might slightly alter behaviors as well, making comparisons even more complex. Thus, due to paucity of field data, we eventually use the assumption that all individual fireflies follow the same IBI distribution.

2) Comparison between the models and the data must be improved, in particular through a quantification of the differences between distributions and sensitivity analysis of the numerical results.

a) Regarding the comparison of the agent-based simulations with experimental data, in Figure 7, we compare the underlying distributions using the two-sided Kolgomorov-Smirnov statistical test for goodness-of-fit. These appear to us the most straightforward and informative approaches, without over-fitting.

b) Regarding sensitivity analysis for the agent-based simulations, for each *β* value from 0 to 1 we statistically compared simulations to the experimental distributions to find the most well-fitted *β*.

c) Finally, owing to experimental constraints leading to sparsity of available data in characterizing the interburst distribution, we strive to strike a delicate balance between sophisticated statistical tools to compare theoretical and simulation distributions (with unrestricted access to large sample sizes) to the finite samples in the empirical distributions. As such, we think it is the apposite to use the first two moments of respective distributions In Figure 3 to show the striking similarity of trends.

3) More discussion of the modeling in connection to past theoretical results and existing literature is necessary to better contextualize the present work and assess its originality.

We have done this closely following the specific suggestions from reviewers.

Revised terminology: removing usage of “model”

Since unintended ambiguity may be caused by use of the word “model”, which could refer to either (1) the theoretical framework, principle of emergent periodicity, and attendant analytic calculation, or (2) the agent-based simulation in the computational realization, we have removed all instances of the word “model” from the results presented in the paper, and replaced by the specific meaning (theory or simulation) in each context.

Similarly, in responding to Reviewers’ comments, we clarify what we understand by their use of the word “model” in each case.

Addressing an error in the agent-based simulation code

We (OM and OP) have now addressed an inadvertent unit typo in the agent-based simulation code. The discharging time (*T_d_*) before the typo was fixed was set to 10000ms. After the fix, the *T_d_* value was correctly set to 100ms. This caused very slow discharges, keeping the voltage high until any β addition was received, resulting in more frequent bursts than we’d actually expect from the model dynamics. This has been fixed, and in our responses to the reviewers, we address the results of this fix by referring to the “unit typo”. We corrected the panels corresponding to agent-based simulation in Figures 3 and 5 to reflect the new numerical simulation results, as well as the corresponding sections in the text of the paper.

Addressing changes to experimental dataset

We increased the size of our N=1 dataset (N is number of fireflies) to correctly match what was reported in the original text of 10 samples. Additionally, we have added characterization of the size of the datasets for N=5, 10, 15, and 20 fireflies.

Reviewer #1 (Recommendations for the authors):The authors propose a simple model for flash dynamics in a certain species of firefly known as P. Carolinus. Remarkably, individual males of this species flash haphazardly, with no particular rhythm, yet in sufficiently large groups, they somehow manage to flash in rhythmic unison. The authors show that their model can account for this phenomenon - with no adjustable parameters - and they test their model quantitatively in idealized experiments on groups of up to 20 fireflies confined to a small, darkened, cuboid tent of dimensions 1.5 x 2 x 1.5 cubic meters.Strengths:The authors' model is unusual in that the individual fireflies are not assumed to be intrinsically rhythmic. By contrast, in most previous work on firefly synchronization the individual fireflies were modeled as "oscillators." That convenient assumption allowed a large body of theory from nonlinear dynamics to be imported. However, for the particular species of firefly being studied here, the authors show the individual males do not flash rhythmically. The authors provide a framework for dealing with this novel case.There are actually two parts to the framework. In the first (extremely stylized) model, the authors assume that after it flashes, each firefly waits a random time before it can flash again. All fireflies choose this random waiting time from the same probability distribution; in that sense, the fireflies are assumed to be statistically identical. Furthermore, the authors assume that when one firefly flashes, it triggers all the rest to flash instantaneously. A weakness of this extreme idealization is that synchrony is thereby built in automatically by assumption, rather than explained as a consequence of the model. But a virtue of this extreme idealization is that it correctly predicts how the variance in the interburst intervals depends on the number of fireflies in the group. In this way, the authors neatly explain a property of the emergent period that they observe in their tent experiments. Most notably, they do this without any adjustable parameters. The result follows as a consequence of their model and their measurements of individual firefly behavior. This is a beautiful instance of using individual behavior to predict collective behavior with the help of some simple, additional assumptions.The second part of the new framework builds on existing research on a class of oscillators known as "integrate and fire" oscillators. The new wrinkle is that the authors introduce a stochastic term in the equation (a random amount of time for the oscillator to charge up) in order to capture the erratic nature of the interburst interval for individual males of this firefly species. The virtue of this more complex model is that it allows the authors to predict a transition to group periodicity as the interaction strength between the fireflies is increased. It also allows the authors to relax the earlier (unrealistic) assumption of instantaneous triggering of all other fireflies whenever one firefly flashes.Weaknesses:The work presented here is an excellent start at understanding the collective behavior of this particular species of firefly. However, the model does not apply to other species in which individual males are intrinsically rhythmic. So the model is less general than it may appear at first.

We take the Reviewer’s point well. We have added text to the paper to clearly highlight this point.

The modeling framework is also developed under the very stylized conditions of experiments conducted in a small tent. While that is a natural place to begin, future work should consider the conditions that fireflies encounter in the wild. Swarms that are spread out in space would require a model with a more complicated structure, perhaps with network connectivity and coupling strengths that both change in time as fireflies move around. This is not so much a weakness of the present work as a call to arms for future research.

We agree with the Reviewer that this is an exciting call to arms for future research!

Overall, the paper does an excellent job of supporting its conclusions with elegant arguments and experiments.This assumption that all individuals have the same IBI distribution could be directly tested. Has this been done? If not, why not? e.g. Are there difficulties with letting one firefly flash long enough to collect sufficient data to fill out the distribution?

1. We have generalized the theory to directly address this point by relaxing the assumption that all individuals exhibit the same inter-burst interval distribution. In short: the main insights continue to hold and we discuss the nuances in the text.

2. Experimentally, hypothesis that all single fireflies isolated from the group exhibit the same interburst interval (IBI) distribution could not be rigorously tested. The main reason is practical: in order to compare IBI distributions across individuals, we would need to collect a large number of fireflies and track them for long durations, which was not realistic given our experimental setup and the short window of firefly emergence. In addition, external environmental factors might slightly alter behaviors as well, making comparisons even more complex. Thus, due to paucity of field data, we eventually use the assumption that all individual fireflies follow the same IBI distribution.

The derivation given in 6.2.1 is clearer than the approach taken here, which unnecessarily introduces Q, q, and c and then never uses them again.

We agree with the Reviewer and have accordingly revised the manuscript.

We have also implemented the suggested edits in the marked up manuscript. We are grateful for the detailed feedback, which helped us substantially extend results, and improve presentation and clarity.

Reviewer #2 (Recommendations for the authors):Synchronous flashing of fireflies is a textbook example of collective behaviour, although much more theoretical work has been devoted to explaining it than field observations. The work proposed in this manuscript, along with previous results by the same group, are significantly contributing to fill the gap between observations and mechanistic models. In these models, single-insect dynamics is described along with the interaction mechanisms.In a clever experiment where fireflies are screened from the rest of the population, so that the number of interacting individuals can be manipulated, the authors were able to characterize both single-fireflies firing patterns, and the emergence of collective-level flashes when their number is progressively increased. These observations show that collective-level firing is more regular and more frequent that the average individual-level firing, a feature that can be explained by globally coupled populations of dynamical systems, where firing is followed by a refractory period. The authors show at first that the increase in coherence (standard deviation of the inter-spike interval of the synchronous flashes) as a function of the number N of fireflies is quantitatively explained knowing just the inter-burst interval of collective flashing at high density - which is largely determined by the duration of the refractory period. Then, by using an integrate-and fire model that explicitly accounts for coupling, they study the dependence of the onset of collective oscillations on coupling, and estimate what coupling strength best explains the observations at different N.I found that the experiment is really nice and has the potential to advance a lot the mechanistic understanding of this collective behaviour. Also, the mathematical model reproduces the main observations, even though density-dependent onset of synchronization, reduction of the oscillation frequency with increasing density, and finite-size scaling are general properties that are observed in populations of globally coupled dynamical systems other than the one proposed here.However, the way models reflect experimental observations and how they compare with them and with one another are in my opinion insufficiently characterized and discussed.I raise two main points:1. The biological relevance of certain hypotheses is insufficiently discussed. This is important because if the observed behaviour is a universal one, alternative models may explain it as well.

We thank the reviewer for raising this point. The main hypotheses underlying our models are: (1) individual fireflies in isolation flash at random intervals; (2) these random intervals are drawn from the empirical distribution reported (implicitly: all fireflies follow the same distribution); (3) once a firefly flashes, it triggers all others. Hypothesis (1) is directly supported by the data presented. Hypothesis (2) is comprehensively addressed in the revised manuscript, as discussed previously. Hypothesis (3) is central to the proposed principle, and enables intrinsically non-oscillating individuals to oscillate periodically when in a group. The resulting phenomenon has been compared to experimental data and extensively discussed in the manuscript. Further, we have also simulated the effect of changing the strength of coupling between fireflies based on this hypothesis in the revised section on agent-based simulation.

2. Comparison between the models and the data could be improved, in particular through quantification of the differences between distributions and sensitivity analysis of the numerical results.

1. Regarding the comparison of the agent-based simulations with experimental data, in Fig. 7, we compare the underlying distributions using the two-sided Kolgomorov-Smirnov statistical test for goodness-of fit. These appear to us the most straightforward and informative approaches, without over-fitting.

2. Regarding sensitivity analysis for the agent-based simulations, for each β value from 0 to 1 we statistically compared simulations to the experimental distributions to find the most well-fitted β.

3. Finally, owing to experimental constraints leading to sparsity of available data in characterizing the interburst distribution, we strive to strike a delicate balance between sophisticated statistical tools to compare theoretical and simulation distributions (with unrestricted access to large sample sizes) to the finite samples in the empirical distributions. As such, we think it is the apposite to use the first two moments of respective distributions In Fig. 3 to show the striking similarity of trends.

I detail below the main points where I think the manuscript could be improved. In general, I think that toning it down a bit would not be out of place. There were also a number of edits that I could signal if I receive a version of the paper with line numbers.A. The assumption that single-firefly spikes obey the same distribution (there is no individual variation in the frequency, or even of the composing number of bursts, of the flash) does not seem to have been verified on the data, that are instead pulled together in one single distribution (Figure 1D). Moreover, the main feature of such distribution is that it has a minimum at 12 secs (discarding the faster bursts that are not considered in the model) and that it is sufficiently skewed so that it takes a minimal coupling for collective synchrony to emerge. I think that the agreement between the distributions for different N would be more meaningfully discussed having previous work as a reference, whereas now this is relegated to the discussion, so that it is unclear how much of the theoretical results are novel and/or unexpected. Quantification of the distance between distributions would also be interesting: it looks like the two models (analytical and simulations) disagree more among themselves than with the data.

Regarding the hypothesis that all individual fireflies exhibit the same interflash interval, please see our response to Main Point 1. Regarding comparing the analytical theory and numerical simulation analysis, Figures 3 and 5 have been revised after a unit typo was found in the code (see Section 2). Following the update, the analytical and numerical models agree in (1) the location of the peak in Figure 3 for all N values, and (2) the peak approaches the minimum of the input distribution as N increases.

B. If I understand correctly, simulations are introduced as a way to get a dependence on the intensity of the coupling (\β). There are several issues here. First, I do not see how the coupling constant could change in the present experimental setup, where all fireflies presumably see each other (different from when there is vegetation). Second, looking at Figure 3, the critical coupling strength appears to depend very weakly from N, and it is not clear how the 'detailed comparison' that leads to the fit is realized (in fact, the fitted \betas look larger that those at which the transition occurs in Figure 3A). I think a sensitivity analysis is needed in order to understand how do results change when \β is changed, and also what is the effect of the natural Tb distribution (Figure 2 F). Results of the simulations might be clearer if instead of using the envelope of the experimental results, the authors tried to fit it to a standard distribution (ex. Poisson) so that it can be regularized. This should allow to trace with higher resolution the boundary between asynchronous and synchronous firing.

We have included agent-based numerical simulations as a way to provide a concrete instantiation of the theory principle and analytical results in the preceding section. While the analytic theory results are fitting parameters free, in the agent-based simulations, we introduce an additional fitting parameter, to see what happens when we relax one hypothesis of the analytical theory: the instantaneous triggering of all fireflies upon an initial flasher. Additionally, the agent-based simulations pave the way for future work, allowing for convenient exploration of the connectivity between individuals and analysis of the behavior of individual fireflies. in this context, please note that Figure 5 has been corrected (see above), leading to a stronger co-dependence of *β* and *N.* In addition to the envelopes, we also report the trends in the first empirical moments (mean and STD) for comparison and tracking of the transition to synchrony.

C. More care should be put in explaining what are the initial conditions hypothesized for the different models. For instance, the results of paragraph 3 are understandable if all fireflies are initialized just after firing, something that is only learnt at the end of the paragraph. I also wonder whether initial conditions may be involved with T_bs in the low-coupling region of Figure 3A not being uniformly distributed, as I would have expected for a desynchronized population.

We have clarified that, indeed, all fireflies are re-initialized after firing. The initial conditions then become a new random vector of interflash intervals. Importantly, we found after receiving the reviews that, due to inconsistent units in our numerical simulation code, Figure 5 was incorrect. With proper units, the new results show a much more widespread distribution at low coupling, as expected by the Reviewer.

D. I found that equations were hard to understand either because one of the variables was not precisely (or at all) defined, or because some information was missing:Equation 1: q is not definedEquation 2: explain what it means: the prob. that others have not flashed times that one flashes. Also, say explicitly what is the 'corresponding PDF.Equation 3: the equation for \epsilon(t) to which this is coupled is missingWhy introduce \β_{i,j} and T_bi if they are then taken independent of the indexes?Definitions of collective and group burst interval should be provided.It would be clearer if t_b0 was defined in the first paragraph of the results, so as to clarify as well its relation with T_b.Define T^i_b in the caption of Figure 3 (they are defined later than the figure is first discussed).The definition of 'the vertical axis label' (maybe find a word for that…) is pretty cumbersome. I could imagine that other definitions would allow the lines in Figure 3 E to converge to the same line for large betas, which would make more sense, considering that in the strong coupling limit I see no reason why the collective spiking should not be the same for different N (the analytical model could help here).

Thank you for these comments; we have incorporated these and related changes.

E. I think that the author's reading of the two 'dynamical quorum sensing' papers they cite is incorrect: De Monte et al. was not about the Kuramoto model, but the same limit cycle oscillators as in Strogatz; Taylor et al. considers excitable systems, potentially closer to noisy integrate-and-fire, at least in that they do not have self-sustained oscillations. Both papers show that oscillations appear above a certain density threshold, and that the frequency of oscillations increases with density, as found in this work. A more accurate link to previous publications in the field of synchronization theory, including the models by Kurths and colleagues for fireflies, would be useful both in the introduction and in the discussion, and would help the reader to position this work and appreciate its original contributions.

1. Thank you for pointing out an inaccuracy in our literature citations regarding synchronization. We have now made corrections to address this point.

2. While we take the Reviewer’s points well, our theory framework (“model”), building off of the principle of emergent periodicity we propose here, is fundamentally different in the nature of individuals from extant “models”. The reference in question has individuals as oscillators, and the fastest frequency is the frequency of the fastest individual oscillator. In contrast, in our work there is no fastest individual oscillator and the “fastest frequency” has a completely different meaning, since individuals do not have a particular frequency associated with them. In this sense, our work is not inspired by theirs. That said, we have included citations as suggested by the Reviewer.

F. The authors say that part of the data is unpublished. I guess they mean that the whole data set will be published with this manuscript. I think the formulation is ambiguous.

Thank you for this comment. We have now clarified that the data will indeed be published with the manuscript.

[Editors' note: further revisions were suggested prior to acceptance, as described below.]

Reviewer #2 (Recommendations for the authors):The authors have significantly improved the manuscript, where assumptions and analytical and numerical results are now presented more clearly.I still have some comments, more of less specific, that I list below, starting with the conceptual ones.1. Citation of previous work on dynamical quorum sensing (lines 51 and 52) I think misses two important points: first these works (and others following them) deal with the appearance of collective oscillations at high density (therefore, the same general problem addressed here); second, Taylor et al. studied also a transition where the oscillators involved did not oscillate at low density, whereas above a density threshold, they display coherent collective oscillations whose period decreases with density – similar to what observed here. I do not think this takes anything away from the originality of this work, which refers to a different system, and models it with different equations, but the parallelism between integrate-and-fire dynamics with quenched noise and excitable dynamics in the presence of noise should in my opinion not be overlooked.

We have explicitly mentioned this in the revised text.

2. As the authors stress in lines 105 and 132, the analytical model shows that all that really matters in this phenomenon is the fastest frequency of the system. This could be used as an argument to say that the actual frequency distribution of individual fireflies is not all that important, as long as their fastest frequency is comparable. The assumption that they are identical would then sound less radical. Ideally, one could use the numerical simulations to check this, as well as the fact that the phenomenon does not break down when the shortest individual interburst interval Tb_min is narrowly distributed (which could also explain why having a few individuals who can flash at a higher frequency does not affect the outcome).

We thank the reviewer for these observations.

3. I still feel that the agreement between the model and observations is a bit overstated (line 120). At least, I think the authors may stress that whereas the model predicts that the frequency of the 7-14 minutes oscillations should increase a lot with N, this is not observed in the data. Maybe this mismatch would be reduced if inter-individual variability was added.

Please see the last three paragraphs of the Discussion section. In reality, as the swarm size increases, we expect that swarms will no longer be all-to-all connected, and the dynamics of the system will depend upon the speed of propagation of information across the swarm. Precisely how this happens is outside of the scope of the current experimental work and theoretical description presented here.

4. In paragraph 4.2, I found it unclear why the authors find it unsurprising that different experiments would correspond to different betas. I think that this point should be discussed, as β and N appear in combination in determining the interaction strength. Otherwise, they could try to fit all distributions with the same β, which would be more natural for me. I guess that the fits would be anyway good to the eye, though quantitatively suboptimal (which could be quantified with the distance introduced).

The reviewer raises valid concerns since as shown in Figure 3, the chosen values for β, the additional fitting parameter introduced in the agent-based simulation, are: *β* = 0.18, 0.13, 0.12 and 0.64 respectively for *N* = 5, 10, 15, 20. We (RS, OM, and OP) find it intriguing that the optimum β clusters around similar values for N = 5, 10, 15, while the optimum β for N = 20 is significantly different. We acknowledge that we do not have an explanation why the fitted parameters values are what they are but note that the fitting curve is flat, implying that several β values could possibly achieve a satisfactory fit. While further agent-based simulations could explore these findings more systematically, we believe that investigating this matter is outside the scope of this paper. Instead, we have acknowledged these points explicitly in the revised discussions.

Portion added to discussions: “As shown in Figure 3, the chosen values for β, the additional fitting parameter introduced in the agent-based simulation, are: *β* = 0.18, 0.13, 0.12 and 0.64 respectively for *N* = 5, 10, 15, 20. Perhaps it is intriguing that the optimum β clusters around similar values for *N* = 5, 10, 15, while the optimum β for *N* = 20 is significantly different. While we do not currently have an explanation for why the fitted parameter values are what they are, we note that the fitting curve is flat, implying that several β values could possibly achieve a satisfactory fit. Further agent-based simulations could explore these findings more systematically, and provide useful insights.”